# Stifled Screams: Experiences of Survivors of Sexual Harassment in First-Generation Universities in Southwestern Nigeria

**Boladale M. Mapayi** [1,2,*], **Ibidunni O. Oloniniyi** [1,2], **Olakunle A. Oginni** [1,2], **Onyedikachi J. Opara** [1],
**Kehinde J. Olukokun** [3], **Abigail Harrison** [4] **and Morenike O. Folayan** [5]

1 Department of Mental Health, Obafemi Awolowo University, Ile-Ife 220101, Nigeria;
  ibiduniyi@oauife.edu.ng (I.O.O.); kaoginni@oauife.edu.ng (O.A.O.); oparakatch@gmail.com (O.J.O.)
2 Mental Health Unit, Obafemi Awolowo University Teaching Hospitals Complex, Ile-Ife 220005, Nigeria
3 Academy for Health Development, Ile Ife 220103, Nigeria; olukokunkehindejoseph@gmail.com
4 Department of Child Dental Health, College of Health Sciences, Obafemi, Awolowo University,
  Ile-Ife 220101, Nigeria; abigail_harrison@brown.edu
5 Department of Behavioural Sciences and Sociology, Brown University, Providence, RI 02903, USA;
  toyinukpong@oauife.edu.ng
* Correspondence: daledosu@oauife.edu.ng

**Abstract:** The aim of this study was to explore the lived experience of survivors of sexual harassment, as well as reportage factors and outcomes, psychosocial sequelae, and how survivors coped in first-generation higher-education institutions in Southwestern Nigeria. A qualitative exploration of the experience of 12 (11 females and 1 male) participants using in-depth interviews was conducted. The findings were grouped into four broad themes, namely: (i) experience of sexual harassment, (ii) reporting patterns, (iii) coping strategies, and (iv) the physical and emotional impact of sexual harassment. Most survivors had experienced sexual harassment multiple times and same-sex harassment occurs in higher-education institutions. Survivors did not report to university authorities because of the perception that the support from the environment was poor. Many shared information with their support networks or visited a psychologist for mental health care. Others used maladaptive coping mechanisms such as increased alcohol consumption. Mental health symptoms ranged from mild to severe. Institutes of higher education need to take decisive actions to improve the environment and to promote the prompt reporting of sexual harassment by survivors; they must also provide access to support to prevent the development of mental health problems, which are a common post-event occurrence, as identified in the present study.

**Keywords:** sexual harassment; Southwestern Nigeria; universities; survivors

## 1. Introduction

Globally, there is a pervasive culture of silence around sexual harassment, especially in higher-education settings, despite it being widespread and a recurring problem (National Academies of Sciences and Medicine 2018). Sexual harassment is often the result of the power imbalance in these settings (National Academies of Sciences and Medicine 2018; Ogunbamero 2006; Imonikhe et al. 2012; Gaba 2010), partly because it is not easy to define (Joseph 2015; Bello 2020). Sexual harassment often includes quid pro quo harassment which is typified by exchanging sex for benefits (performance or academic favors), and or to avoid some form of disadvantage (e.g., a demotion at work or academic failure). It is often perpetrated in a hostile environment, where speech or conduct creates an intimidating or humiliating environment that negatively affects an individual's academic or job performance (MacKinnon and MacKinnon 1979; Equal Employment Opportunity Commission 1997).

Sexual harassment is often associated with physical and verbal assault, emotional violence, bullying, coercion, discrimination, and intimidation, which often affect the physical and emotional health of the survivor (National Academies of Sciences and Medicine 2018).

These effects include suboptimal academic fulfillment resulting from absenteeism, changing courses or advisors, and even leaving school entirely (Huerta et al. 2006; Fitzgerald 1990); psychological distress with symptoms of depression, stress, and anxiety, self-blame, lowered self-esteem and impaired psychological well-being (Bond et al. 2004; Cortina et al. 2002; Langhout et al. 2005; Lim and Cortina 2005; Taiwo et al. 2014); and physical health issues such as headaches, exhaustion, sleep problems, gastric problems, nausea, respiratory complaints, musculoskeletal pain, and weight loss or gain (De Haas et al. 2009; Wasti et al. 2000). Sexual harassment committed by a superior is sometimes more harmful than peer-perpetrated harassment (O'Connell and Korabik 2000).

Sexual harassment in colleges and universities is grossly underreported (Joseph 2015). This is often because of unequal power relations, a fear of the loss of one's status, marks, or job as retaliation, and the attendant stigma that it brings. Survivors cope with sexual harassment in many ways including ignoring or appeasing the harasser or seeking social support (National Academies of Sciences and Medicine 2018). Higher rates of sexual harassment are reported in environments that tolerate sexual harassment, as is reflected by the absence of institutional responses to complaints, failure to sanction perpetrators, and protecting complainants from retaliation (National Academies of Sciences and Medicine 2018). The silence around sexual harassment is widespread in higher-education institutions in Nigeria. Several studies have investigated its prevalence and associated factors in Nigeria (Owoaje and Olusola-Taiwo 2010; Okeke et al. 2021) but few have explored the reasons why sexual harassment is underreported, what can be done to improve reporting, and the lived experience of survivors.

The present study recognizes that sexual harassment is a multidimensional problem. This study was therefore framed using multiple theoretical frameworks to identify the factors associated with the reporting of sexual harassment by students in higher-education institutions in Nigeria. The sex role spillover theory, which highlights the carryover of gender-based roles into workspaces, explains why the biological and socio-cultural dominance of men over women in societies makes the education space more likely to be accepting of men (as superiors) responding sexually towards women (Gutek and Morasch 1982). The hierarchical structure of the higher institutions entrenches power and authority relations, which implicitly encourage harassment based on embedded power relations between men and women, as highlighted by the organizational theory (Gruber 1992). This power imbalance can also be extended to the lecturer–student relationship. Sociocultural theory highlights that the socialization of men into roles of sexual assertion and women as submissive makes sexual harassment a way for men to express dominance; hence, they are more likely to be the perpetrators and females the most likely victims (Kitzinger and Thomas 1997). This present study, however, conceptualized survivors and perpetrators based on the unequal power relations between male and female students and also between students and lecturers.

There is, however, a lack of a conceptual framework for the study of sexual harassment responses and coping mechanisms, which constitute a multidimensional construct that cannot be represented by a single continuum. We recognize that responses vary from self-focused responses (coping strategies that do not involve the perpetrator) to initiator-focused responses (addressing the perpetrator and the event directly, such as by reporting the case) (Knapp et al. 1997). The present study explored both forms of response and assessed the impact of the forms of response on the social and mental health sequelae of sexual harassment among survivors.

The present study aimed to investigate the experience of sexual harassment by men and women in heterosexual and same-sex situations in first-generation universities in South West Nigeria. We explored the lived experience of SH survivors, the social and mental health sequelae of sexual harassment among survivors, factors associated with reportage, the outcome of reportage, and how survivors coped and their recommendations for their institutions.

## 2. Materials and Methods

The present study is part of a larger mixed-method study (Mapayi et al. 2023) that was conducted in three Nigerian universities to explore the perceptions that drive heterosexual and same-sex sexual harassment among students and staff, the institutional mechanisms that exist to prevent and respond to sexual harassment, the social and mental health consequences among survivors, determinants of the decision to report, the resultant actions taken, and the lived experience of survivors.

Study design: the present report is a qualitative study using in-depth interviews (IDIs) to explore the reporting of sexual harassment, coping strategies, and the impact of sexual harassment on the social and mental health of student survivors who experienced sexual harassment in first-generation higher-education institutions in South West Nigeria, where the culture is largely homogenous.

Study sites: the study sites were the oldest public universities that were established between the end of the 1940s and the beginning of the 1960s. These were the University of Ibadan, the Obafemi Awolowo University, and the University of Lagos (Statista 2018). These universities have a large student and staff population from all the major tribes in Nigeria.

Sample: the target population for this study comprised students of the selected universities. Only students who were above 18 years old, who had survived sexual harassment, and who gave consent for study participation were recruited for the study. Students who were physically or emotionally ill and could not participate in the study were excluded. Students who consented took part in this research study between August and December 2022. Participants were able to read and communicate in the English language.

Procedure: a purposive participant selection approach was adopted to identify and recruit survivors. Recruitment was conducted through interactions with the core units of each of the three universities (the Directorate of Student Affairs, counseling, security, personnel units, and centers for gender studies) that have had interactions with sexual harassment survivors. Key officers in these units interacted with the survivors and introduced the study to them, and interested study volunteers were referred to the study team. Participants consented, and the medium for the conduct of the interviews (face-to-face interview, telephone interview, or video interview), as well as the day and time for the IDI, was chosen by the study participants. The interviews explored the experience and the consequences of sexual harassment by the survivor, whether the event was reported or not, the outcomes of either decision, the perception of the outcome, and how the survivor coped. The interview process ensured privacy, confidentiality, and the participants' freedom to share their experiences without reservation. Research assistants were trained on how to offer psychological first aid to survivors as well as psychological support to cope with vicarious trauma.

Instruments: the interview guide used for the present study was developed by the authors after a desk review of similar studies in the global literature. This guide was shared with survivors and other stakeholders during the community entry stakeholders' meetings for their inputs and revisions. The guide was then revised with input from survivors before use.

Data analysis: analysis of the interviews commenced with the verbatim transcription of the audio recordings. The accuracy, integrity, and completeness of all transcriptions were verified by passing the transcripts through 2-level proofreading. At each level, the recordings were listened to and read along with the typed transcripts to ensure all transcription discrepancies were corrected. Thereafter, all transcripts were coded by three experts, a thematic analysis was conducted using NVivo 12 Pro, and areas of divergence were resolved through discussions between the study leads (M.B.M., O.I.O., and O.J.O.). Deductive analysis was conducted, and the findings were synthesized and presented in response to the research questions.

Ethical considerations: the research protocol was submitted for ethical review and approval received from the Institute of Public Health Research Ethics Committee (IPH/OAU/12/2028), Obafemi Awolowo University; the University of Ibadan Health Research Ethics Committee (UI/EC/22/0313); and the University of Lagos Health Research Ethics Com-

mittee (CMUL/HREC/08/22/1082). Written consent was obtained from the participants. Confidentiality was ensured and psychological first aid was offered to all participants. Referrals for therapy were made for participant who exhibited or reported distress.

## 3. Results

### 3.1. Background Information

Four IDIs were conducted in each of the three institutions. In all, 18 people were approached at the three universities. Two did not meet the inclusion criteria (they were not students when the sexual harassment occurred) and four declined to participate. The non-responders were mostly sexual minority individuals who had been sexually harassed by male lecturers and students. One female respondent declined because she was not ready to tell her story.

The ages of the 12 participants that consented and were interviewed at the time the sexual harassment occurred ranged between 18 and 25 years. All survivors were students and ten survivors knew the perpetrator. There were 11 female participants and 1 male participant. The forms of the sexual harassment experienced ranged from physical touch to assault, attempted rape, and rape. The locations of the incidents included the perpetrator's abode or office, a lecture room/theatre, the school library, and a party (Table 1). The results are grouped into four broad themes, namely, (i) the experience of sexual harassment, (ii) reporting patterns, (iii) coping strategies, and (iv) the physical and emotional impact of sexual harassment.

### 3.2. Experience of Sexual Harassment

For this study, the experience of SH ranged from physical touch to assault, attempted rape, and rape. The perpetrators of sexual harassment ranged from students (n = 6) to lecturers (n = 5) and non-academic staff (n = 1). Half of the survivors (n = 6) had experienced sexual harassment multiple times.

*"This was in a public area, the library, we were just talking, and then all of a sudden, he just grabbed me and started touching me in sensitive parts. And I was telling him to stop but he kept on doing it."* (Female survivor 1 Uni 3)

*"I think it has created a pattern where in sexual experiences I tend to be taken advantage of generally and I think that's what has led to this repetitive sexual assault."* (Male survivor 2 Uni 1)

### 3.3. Reporting Pattern of Sexual Harassment

The formal reporting of cases of sexual harassment was poor. Only three of the survivors reported the event to the school authorities, while none reported it to a law-enforcement agency. Reasons for not formally reporting ranged from avoiding stigma (n = 7), the feelings of internal self-blame, guilt, and shame (n = 5), to poor handling of previously reported cases by the school authorities (n = 4), to the protection of the perpetrators (n = 3), and the highly influential positions of the perpetrator's families (n = 3). Further reasons for not reporting include a perception of not having enough proof (n = 3); the fear of negative repercussions following reportage (n = 3); being begged not to report (n = 2); unsupportive staff (n = 1); and past negative experiences with law-enforcement agents (n = 1) (Table 2).

*"When the lecturers came and asked me what happened and I told them they were begging me to leave him alone when they heard that he was a student with a carryover . . . . You know. people that are even much older than my mum was kneeling for me. I did not have any choice."* (Female survivor 5 Uni 2)

*"The last time I tried to report to law enforcement, they were like Oh, wow when they saw blood all over me. The moment the guy appeared and they know he is a cultist they were like I should better run away for my life."* (Female survivor 5 Uni 2)

" . . . I feel like it is my fault and there is something that I could have done to change it but I did not . . . . so, I just have to just leave it and live with the pain and guilt." (Female survivor 11 Uni 3)

One survivor who experienced same-sex sexual harassment did not report it to protect the identity and reputation of the sexual minority community and the perpetrator.

"I felt it was homosexual, and was also protecting him and preserving the person's reputation. I know how hard it is for non-traditional men when it comes to sexuality and so, I just didn't take it up." (Male survivor 2 Uni 1)

### 3.4. Coping Strategies for Sexual Harassment

The identified coping strategies for sexual harassment by survivors were categorized into normalization, engagement, help-seeking, and detachment. Normalization encompasses acceptance, denial, refusal, grief, silence, and tolerance. Engagement involves confrontation, negotiation, retaliation/threatening, and discrimination of the perpetrators. Help-seeking involves discussing with friends, complaining to school authorities, consulting professionals, and confronting perpetrators. Lastly, detachment includes withdrawal, distancing, or leaving school (Worke et al. 2021). Some of these strategies can be maladaptive, with significant negative impacts on the lives of survivors (Ford and Ivancic 2020). According to the current study, survivors used several coping mechanisms ranging from talking with peers, joining a church community and receiving support from the church, joining social clubs, and heavy drinking; two survivors sought professional help. One survivor developed a risky fetish.

"I started drinking a lot. I drank heavily around that period. The alcohol was temporary . . . . for that moment you forget . . . ." (Female survivor 4 Uni 1)

"I joined like my church community . . . I was able to tell my pastor (hmm) and I really like the way he's handling it . . . " (Female survivor 1 Uni 1)

"Then I talked to a therapist . . . a psychiatrist . . . I just had to . . . because it was just messing with my head." (Female survivor 4 Uni 1)

"I think at some point, I felt it was a badge of honor to have a sexual encounter with a lecturer, but I started realizing how problematic that is. I had started fetishizing power play being at the receiving end of the entire thing and people using their power to get sexual favors from me. I started fetishizing that." (Male survivor 2 Uni1)

Most of the survivors (n = 9) shared their experiences of sexual harassment with close friends, roommates, and family members. The three survivors who had not disclosed their experience to anyone before the interview were concerned about being blamed and judged by others.

### 3.5. The Physical and Emotional Impacts of Sexual Harassment

Participants in the current study felt a range of emotions following the sexual harassment. Half (n = 6) felt sad, depressed, or low; five felt bad, awful, terrible, traumatized, or horrible; and four reported crying and being teary. Other emotions that were felt include feeling used, angry, and guilty. The majority (n = 9) became suspicious of men or people, stopped trusting men/people, were scared of men, or did not feel safe around men. Some had suicidal thoughts and flashbacks of the event, or became clinically depressed.

"I feel like it's becoming a problem for me because, at this age, I don't want to be emotionally attached to anybody, any male, My Twitter account was banned because a lot of people reported my account because I always talk about my hate for men. I just don't see that gender as a gender I want to like spend the rest of my life with, I feel I'm better off being alone." (Female survivor 1 Uni 1)

"Instead of me being totally disgusted, I think it made me want to have other sexual experiences with maybe someone that was also a staff and maybe older than me. I think it kind of intensified the reason to be submissive in a power game." (Male Survivor 2 Uni 1)

**Table 1.** Background information on survivors of sexual harassment.

| S/No | Code | Sex | Age | Age at SH | Level at SH | Type of SH | Sex of Perp | Role of Perp | Location of SH | Emotions after SH | Tell Anyone? | Psychosocial Sequelae | Reportage | Reasons for Not Reporting | Coping Mechanism |
|---|---|---|---|---|---|---|---|---|---|---|---|---|---|---|---|
| 1. | Surv1/Uni1 | Female | 22 | 19 | 100 L | Tapped her buttocks | Male | Lecturer | Lecturer's apartment | Self-blame | No | Suicidal thoughts | No | Fear, people will blame me | My church community |
| 2. | Surv2/Uni1 | Male | 21 | 20 | 200 L | Rape | Male | Lecturer | Lecturer's apartment | Felt dirty, devalued | Yes, friends | Suspicious of people | No | Protecting the perpetrator | Talked to friends |
| 3. | Surv3/Uni1 | Female | 19 | 17 | 100 L | Rape | Male | Student | Perp's apartment | Felt used | Yes, parents | Stopped trusting men | Yes—school | Reported | Avoidance |
| 4. | Surv4/Uni1 | Female | 23 | 20 | 400 L | Rape | Male | Lecturer | Perp's office | Betrayed, dirty | Yes, friend | Flashbacks, depression, Suspicion | No | Perpetrator was 'untouchable' | heavy alcohol drinking |
| 5. | Surv5/Uni2 | Female | 22 | 20 | 100 L | Attempted rape | Male | Student | Lecture room | Felt low, Cried | Yes, Her friends | Does not trust people | No | Begged not to report | heavy alcohol drinking |
| 6. | Surv6/Uni2 | Female | 22 | 19 | 200 L | Groped | Male | Student | Lecture theatre | Surprisedfelt awful | Yes, room mate | Stopped trusting men | No | Lack of evidence | Joined 'Kegite club' |
| 7. | Surv7/Uni2 | Female | 26 | 25 | 100 L | Rape | Male | Staff | Perp's apartment | traumatized, bad | Yes, her friends | Suspicious of people | No | Fear of effect on reputation | Plays music, Avoidance |
| 8. | Surv8/Uni2 | Female | 25 | 25 | 200 L | Rape | Male | Lecturer | Lecturer's office | Felt depressed | No | People talking about her | Yes—school | Reported | A CSO Prayers |
| 9. | Surv9/Uni3 | Female | 22 | 21 | 200 L | Grabbed her buttock | Male | Student | A party | Sad, angry | Yes, friends | Flashbacks | Yes—school | Reported | |
| 10. | Surv10/Uni3 | Female | 21 | 17 | 100 L | Attempted rape | Male | Student | Perp's apartment | Felt really bad, teary | Yes, close friends | Stopped trusting men | No | Perpetrator was 'untouchable' | Support from friends |
| 11. | Surv11/Uni3 | Female | 20 | 19 | 100 L | Forceful kissing, groping | Male | Student | Library | Felt guilty, and teary | Yes, a friend | Scared of men | No | Feared others' judgment | Support from friends |
| 12. | Surv12/Uni3 | Female | 20 | 20 | 200 L | Attempted rape | Male | Student | Perp's hostel | Teary and felt guilty | No | Suspicious of everyone | No | Feared others' judgment | Crying |

**Table 2.** Reasons for not reporting the experience of sexual harassment.

| S/No | Reasons for Not Reporting Sexual Harassment | Surv1/Uni1 | Surv2/Uni1 | Surv4/Uni1 | Surv5/Uni2 | Surv6/Uni2 | Surv7/Uni2 | Surv10/Uni3 | Surv11/Uni3 | Surv12/Uni3 |
|---|---|---|---|---|---|---|---|---|---|---|
| 1. | Stigma | ✓ | | | ✓ | | ✓ | ✓ | ✓ | |
| 2. | Self-blame, guilt, and shame | ✓ | | | | ✓ | | ✓ | ✓ | ✓ |
| 3. | Outcome of previous cases | ✓ | | ✓ | | | | ✓ | ✓ | |
| 4. | Lack of evidence | ✓ | | ✓ | | ✓ | | | | |
| 5. | Lack of courage to report or fear of a negative repercussion | ✓ | | ✓ | | | | | | ✓ |
| 6. | The belief that the system would protect the perpetrators | | | ✓ | | | | ✓ | | |
| 7. | The perpetrator's family is highly influential | | | ✓ | | | | ✓ | | |
| 8. | People not believing them over the perpetrator | ✓ | | ✓ | | | | | | |
| 9. | People begging them not to report | | | | ✓ | | | | | |
| 10. | Not wanting their parents to find out | | | | | | | | ✓ | ✓ |
| 11. | Protecting the identity of the perpetrator | | ✓ | | | | | | | |
| 12. | Previous encounters with law enforcement agents | | | | ✓ | | | | | |
| 13. | Took pride in the encounter | | ✓ | | | | | | | |
| 14. | The attitude of other members of staff | ✓ | | | | | | | | |

## 4. Discussion

This is the first study conducted in Nigeria to explore the experiences of survivors of sexual harassment in higher institutions, despite the multiple reports and publications on the occurrence of this phenomenon (Akpotor 2013; Bello 2020; Imonikhe et al. 2012; Ladebo and Shopeju 2004; Ogunbamero 2006; Okeke et al. 2021; Omorogiuwa 2018; Onasoga et al. 2019; Suleiman 2017; Taiwo et al. 2014). This study found that survivors had experienced sexual harassment from diverse community members of higher-education institutions; most participants do not report cases of sexual harassment; both adaptive and maladaptive coping strategies are used; and most survivors experience short- and long-term physical and mental health difficulties associated with their experience of sexual harassment.

Similarly, to the findings of other studies (Rosenthal et al. 2016; Wood et al. 2021), the survivors we interviewed had experienced sexual harassment perpetrated by students and academic and non-academic staff members of their institution. Additionally, as was the case in other studies, (Cantor et al. 2015; Sivertsen et al. 2019), sexual harassment took many forms including unwanted physical contact, sexual advances, sexual comments, and rape.

Unlike any prior report in the literature on sexual harassment in Nigeria, our study determined that sexual minority individuals also experience sexual harassment from faculty members. This is an under-explored area of research in Nigeria, where there is a culture of silence not only around rape but also around sexual identity (Makanjuola et al. 2018; Ojoniyi 2018). The stigma and the physical and emotional consequences of openly identifying as a gender minority individual in Nigeria increases the risk of the non-reporting of sexual harassment by community members. Same-sex relationships are criminalized in Nigeria (Adamu 2019) and some survivors may struggle with their sexual expression and report greater sexuality-related stigma after the experience (Davies 2002). These factors may impede the survivor from reporting the incident and seeking support, thereby increasing the risk for the experience to have a long-term mental health impact. A past study indicated that sexual minority individuals were more likely than straight individuals to report sexual harassment and sexual abuse (Smith et al. 2022). The reasons for their high risk for sexual harassment are poorly understood, though they may be related to homophobic attacks in the form of harassment (Lauckner et al. 2019). Our study did not explore this context either. However, we noted that some other survivors and the sexual minority interviewee had experienced sexual harassment multiple times.

The immediate management of all sexual harassment survivors may reduce the health impact of sexual harassment. We observed that some survivors sought support from friends and family, counseling services and support groups, engaged in self-care activities, or had sought legal action in the past. These actions were all self-driven and not supported by the institutions where the event took place. While it may have been possible for the institution to offer therapeutic support, it appears that the environment in higher-education institutions in Nigeria makes access to and uptake of such services difficult for survivors (Behre 2017; Linder and Myers 2018). Where access to therapeutic support is limited, as it may be in the case of sexual minority individuals, individuals may resort to the adoption of maladaptive coping strategies such as the use of psychoactive substances that can increase the risk for further sexual harassment (Oginni et al. 2019; Filipas and Ullman 2006).

When sexual harassment happens between peers, the perpetrator is often perceived to have greater personal power, and there was a repeated history of harassment. Most survivors of sexual harassment in schools had had multiple such experiences (Rapidah et al. 2017). The perceived causes of the repeated experience of sexual harassment are keeping silent and not reporting the offenders, the use of alcohol and drugs, the portrayal of women as sex objects in the media, inadequate security on campus, and indecent dressing (Onasoga et al. 2019). Generally, individuals who have experienced sexual violence or trauma in the past may be more vulnerable to sexual harassment (Follette et al. 1996). This can be due to feelings of shame and guilt, or a lack of trust in others (Vidal and Petrak 2007). They may also utilize risky behaviors, such as substance use, which can increase the likelihood of sexual harassment, as they can impair a person's judgment and ability to protect themselves

from unwanted advances (Caron and Mitchell 2022). The repeated nature of sexual harassment can make it particularly difficult for victims to address or report, as they may feel intimidated or fear retaliation (Wright and Bean 1993). Repeated sexual harassment can create a toxic and hostile environment, affecting not only the victim but also their ability to learn, work, or participate in social activities (Schneider et al. 1997; National Academies of Sciences and Medicine 2018). It can lead to feelings of isolation, anxiety, depression, and post-traumatic stress disorder (PTSD) among victims (Mason and Lodrick 2013).

Sexual harassment is associated with an increased risk of anxiety, depression, and post-traumatic stress disorder, as well as diminished self-esteem, self-confidence, and psychological well-being (Okeke et al. 2021). Some survivors may also withdraw from school, have worsening school grades, or start to engage in risky sexual behaviors (Kaltiala-Heino et al. 2018). As depicted by the results of this study, sexual harassment survivors reported experiencing all the listed emotional and mental health challenges inclusive of academic difficulties and physical health problems (Duffy et al. 2004; Huerta et al. 2006; Cortina et al. 2002; Langhout et al. 2005; Lim and Cortina 2005; Taiwo et al. 2014). Considering that not all participants demonstrated mental health difficulties, these mental health difficulties may be more pronounced in those with pre-existing genetic risks (Manuck and McCaffery 2014). Alternatively, given the stigma associated with mental health disorders (Adewuya and Makanjuola 2008), some participants may have been unwilling to disclose these difficulties during the interview. Some survivors utilized maladaptive coping strategies such as drinking alcohol excessively as a means of dealing with the trauma experienced, while others found solace in the communities around them and received positive support from those communities. This is viable option for survivors, which can be strengthened by raising awareness in all communities (Holland and Cortina 2017; Scarduzio et al. 2018).

There are multiple reports on the silence around sexual harassment and sexual abuse in Nigeria (Ajuwon and Adegbite 2008; Akoja and Anjorin 2020; Awaah 2019). While the #MeToo movement had tremendous impacts in the US and parts of Europe (Hillstrom 2018; Sweeny 2020), the same cannot be said for other parts of Europe (Gersen 2017) and Nigeria, where the culture of silence is still rife. As indicated in the present study, the culture of silence is heightened when perpetrators are spared the physical fear of being prosecuted and survivors are begged to remain silent, their 'screams stifled into silence', and they live with the invisible scars of emotional trauma. Other reasons that were identified earlier are the fear of retaliation, the fear of not being believed, the fear of being stigmatized, shame, blame, and embarrassment, a lack of trust in the institution's ability to be just and fair, the belief that nothing will change, the feeling of helplessness and a lack of agency, the desire for privacy, concerns about one's career or academic advancement, trauma, and emotional distress (Spencer et al. 2017; Alaggia and Wang 2020; Caron and Mitchell 2022).

The limitations of this study included the presence of the researcher during the data gathering, considering the sensitive nature of the subject; however, this was mitigated by offering different platforms for the interviews and, indeed, all survivors opted for the virtual interface as opposed to face-to-face interviews. Another limitation is that, because of the issues around same-sex relationship criminalization in Nigeria, a number of same-sex sexual harassment survivors declined to participate in the study.

The implications of the findings from the present study call for universities to improve their support resources for survivors of sexual harassment. The need to educate the campus community on ways to offer support during and after disclosures is essential, as many survivors seek help from the community around them. Institutions must create a safe and supportive environment where victims feel empowered to report harassment and receive appropriate support and care, regardless of whether they choose to make a formal report (Smith and Freyd 2014). Hirsch and Khan (2020) offered an extensive discussion of the tortuous journey for survivors within a Columbia University in the book *Sexual Citizens*. At the same time, advocates working on sexual harassment should invest resources in increasing public awareness of the physical and emotional disorders associated with sexual harassment and improve the information for survivors on how to manage experiences

when they are 'screamed into silence' by multiple environmental factors that make the reporting of cases difficult. Future studies are needed to explore the trajectories of survivors who used positive coping strategies so that such successful approaches may be scaled up.

## 5. Conclusions

Sexual harassment occurs in higher-education institutions in Southwest Nigeria; survivors often do not report these incidents because of the perceived poor environmental support to do so. Moreover, survivors adopt adaptive and maladaptive coping mechanisms to deal with the emotional challenges associated with the experience of sexual harassment. Same-sex harassment also occurs in higher-education institutions in Nigeria. Future studies are needed to explore the factors that enable survivors to adopt adaptive rather than maladaptive coping strategies when dealing with sexual harassment, and efforts should be made to scale up these self-care-promoting factors in Nigerian higher-education institutions. Self-reports that delve into gender dynamics and sexual harassment may also help us to understand issues around repressed sexuality due to bans on same-sex relationships.

**Author Contributions:** Conceptualization, B.M.M. and I.O.O.; methodology, B.M.M. and I.O.O.; software, I.O.O.; validation, B.M.M., O.A.O. and I.O.O.; formal analysis, I.O.O.; investigation, O.A.O. and K.J.O.; resources, B.M.M. and M.O.F.; data curation, I.O.O.; writing—original draft preparation, B.M.M.; writing—review and editing, B.M.M., M.O.F., I.O.O., O.A.O.; visualization, B.M.M.; supervision, M.O.F. and A.H.; project administration, O.J.O.; funding acquisition, B.M.M. All authors have read and agreed to the published version of the manuscript.

**Funding:** This research was funded by the Consortium for Advanced Research Training in Africa (CARTA), grant number CARTA/2022/002, and the APC was funded by CARTA. This research was supported by the Consortium for Advanced Research Training in Africa (CARTA). CARTA is jointly led by the African Population and Health Research Center and the University of the Witwatersrand and funded by the Carnegie Corporation of New York (Grant No. G-19-57145), Sida (Grant No: 54100113), the Uppsala Monitoring Center, Norwegian Agency for Development Cooperation (Norad), and by the Wellcome Trust [reference no. 107768/Z/15/Z] and the UK Foreign, Commonwealth and Development Office, with support from the Developing Excellence in Leadership, Training and Science in Africa (DELTAS Africa) programme. The statements made and views expressed are solely the responsibility of the Fellow. For the purpose of open access, the author has applied a CC BY public copyright license to any author-accepted manuscript arising from this submission.

**Institutional Review Board Statement:** The study was conducted in accordance with the Declaration of Helsinki and approved by the Institutional Review Board (or Research Ethics Committee) of the Institute of Public Health, Obafemi Awolowo University (IPH/OAU/12/2028; 24 August 2022); the University of Ibadan Health Research Ethics Committee (UI/EC/22/0313; 21 September 2022); and the University of Lagos Health Research Ethics Committee (CMUL/HREC/08/22/1082; 15 September 2022).

**Informed Consent Statement:** Informed consent was obtained from all subjects involved in the study.

**Data Availability Statement:** The data presented in this study are available on request from the corresponding author. The data are not publicly available due to privacy and ethical restrictions.

**Acknowledgments:** We acknowledge the support of all the survivors who shared their stories.

**Conflicts of Interest:** The authors declare no conflict of interest. The funders had no role in the design of the study; in the collection, analyses, or interpretation of data; in the writing of the manuscript; or in the decision to publish the results.

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
