# Peer review of "Stifled Screams: Experiences of Survivors of Sexual Harassment in First-Generation Universities in Southwestern Nigeria"

_socsci, doi:10.3390/socsci12070401_

Round 1
Author Response
Dear Reviewer, many thanks for your kind review of our manuscript. Please find our comments attached.

Reviewer 2 Report
Two particular issues I found which can be easily remedied. More specific recommendations for future research - meaning with the discussion on repressed sexuality and issues due to ban on same sex relationships - it might be recommended to have a self-report to specify more on gender dynamics and sexual harassment. It may also be helpful to look at recent literature post pandemic with students adjusting to issues regarding sexual harassment now back in classrooms. Many were exposed to harassment, abuse and assault while trapped with abusers.
The second is minor but thoughts to maybe discuss and insert paragraph on #MeToo and its inability to have standing among Nigerian universities. Meaning, the movement had a tremendous impact in US and parts of Europe, however countries like France, UK still dealing with disproportioned levels for students, graduate students and tenure trach professors. Just an idea to tie in generalizability with university issues.
Author Response
Dear Reviewer, thank you for your kind review of our manuscript. Please find attached our comments.
